# SWAN Modeling of Dredging Effect on the Oued Sebou Estuary

**Nisrine Iouzzi [1,\*], Laila Mouakkir [1], Mouldi Ben Meftah [2,\*]**  **, Mohamed Chagdali [1] and Dalila Loudyi [3]**

1 Faculty of Sciences Ben M'Sik, Hassan II University of Casablanca, Avenue Driss El Harti, Sidi Othmane-Casablanca BP 7955, Morocco
2 Department of Civil, Environmental, Land, Building Engineering and Chemistry, Polytechnic University of Bari, Via E. Orabona 4, 70125 Bari, Italy
3 Faculty of Sciences and Techniques, Hassan II University of Casablanca, Mohammedia BP 146, Morocco
\* Correspondence: iouzzi.nisrine@gmail.com (N.I.); mouldi.benmeftah@poliba.it (M.B.M.); Tel.: +39-080-5963508 (M.B.M.)

**Abstract:** The estuary ecosystem's health and ecological integrity are essential for preserving environmental quality, habitats, and economic activity. The main objective of the present study is to comprehend the wave hydrodynamic impact on the Oued Sebou estuary, which is situated in the Kenitra region on Morocco's north Atlantic coast in North Africa. Specifically, it focused on the dredging effect (caused by sand extraction) on the wave motion and its impact on the estuary environment. Different scenarios of wave-propagation simulations were carried out, varying the significant wave height, in deep water (from 1.5 to 4 m), and considering the bathymetry before and after two dredging cases of 2- and 4-m depths. The change of wave height at the Oued Sebou estuary shoreline was simulated by using the third version of the Simulating Waves Nearshore Model (SWAN). The SWAN model formulates the wave evolution in terms of a spectral energy balance on a structured grid. The effect of dredging on the wave spreading in addition to the flow hydrodynamic structures were extensively analyzed. According to the simulated results, the dredging activities in the Oued Sebou estuary mainly affect the river mouth and the southern breakwater area, increasing the potential erosive action. The areas at the northern coastal strip and near the northern breakwater are subject to possible accumulation of sediments.

**Keywords:** Oued Sebou estuary; bathymetry; dredging; SWAN; wave spreading; flow hydrodynamic structures

## 1. Introduction

Estuaries are one of the most important interconnections between land and sea. They are frequently important areas for leisure and economic activities. Estuaries are also ecosystems that are very susceptible to pollution and environmental disturbances and must be adequately preserved. In estuaries, rivers and oceans interact through a number of intricate phenomena: exchange of water of different densities, sediments, pollutants, nutrients, organic matter, and biota. Therefore, it is necessary to have a better understanding of the hydrodynamics of estuaries and coasts in order to effectively assess and monitor the environment quality in these regions and to predict coastal evolution [1]. It takes appropriate methodologies combining theoretical analysis and modeling studies to anticipate the hydrodynamics of estuaries and coasts, which are caused by complicated mechanisms linking mass exchange to heat transfer processes [1].

Recently, several research studies were conducted to investigate the hydrodynamics of estuaries [2–12]. Many driving conditions, such as river flow characteristics, tidal amplitude, sediment properties, surges, waves, currents, and bathymetry, affect the dominant physical processes in an estuary. The primary sources of energy input into estuaries are typically tides, surges, and waves. Wave dispersion in shallow water is significantly impacted by the change in bathymetry, leading to refraction, diffraction, reflection, and shoaling. Natural

beaches and manmade coastal structures that reflect waves can have a significant impact on the hydrodynamic structures and, consequently, the transport of sediment in front of the reflector [13]. According to Chang and Hsu [13], the prediction of wave reflection coefficients is still a challenging task. Before marine structures are created, wave reflection at them is typically evaluated with physical models for engineering practice. Incident waves are also reflected by the sloping bathymetry. The drag coefficient estimation is complicated by the superposition of incident and reflected waves (due to the variation in bathymetry and the existence of any natural or man-made structures). In their study, Chang and Hsu [13] compiled many approaches for predicting the reflection coefficient of the waves suggested in the literature and highlighted their limitations. Most of these methods, according to the authors, are ineffective at predicting the reflection coefficients of waves on a bed with arbitrary or natural bathymetry. Based on a linear wave shoaling theory (without considering wave breaking), determining changes in wave height and phase due to bathymetry variations, Chang and Hsu [13] proposed a simple frequency-domain method for separating incident and reflected waves to account for normally incident linear waves propagating on an inclined bed with arbitrary 2D bathymetry. According to the authors, their method is applicable to both laboratory and field conditions predominantly for normal shores on which deep-water waves are propagated.

In intermediate and shallow waters, waves from deep waters begin to interact with the seabed, undergoing various transformations. These transitions are the results of many phenomena, for example, refraction, diffraction, reflection, shoaling, breaking, and swash. Wave prediction is of crucial importance to human activities as it provides useful information for many coastal engineering applications such as the coastal protection, environmental monitoring, navigation, safe port management, and good control of recreational activities in popular coastal areas [14]. Both observational and measurement data, as well as physical and numerical models, are often used to assess extreme marine events around the world [15]. Chi and Rong [15] confirmed that long-term in situ observations can accurately estimate the sea level variation but are usually spatially limited. The authors also indicated that many numerical models have been developed and validated to reproduce the spatial and temporal variation of wave spreading and uncover the underlying dynamics. The sea wave motion is strongly nonlinear and greatly influenced by many factors, i.e., the seabed, wind velocity, current circulations, and induced radiation stress. The interaction of the waves with different factors leads to complex hydrodynamic behaviors, making their numerical simulation very uncertain, which always requires validations using measured data.

The Simulating Waves Nearshore (SWAN) model is one of the most very popular wind–wave models [16–27], used by many government organizations, research institutes and consulting companies worldwide, to predict the size and forces of waves, allowing for changes in wave propagation from deep water to the surf zone [14]. Based on SWAN manuals and as outlined in Lin [14], the model's primary function is to resolve the spectral action balance equations, which represent the effects of spatial propagation, refraction, shoaling, generation, dissipation and nonlinear wave–wave interactions, without imposing any a priori constraints on the spectral evolution of wind waves. Hoque et al. [16] applied the SWAN model to forecast storm wave conditions in the shallow nearshore region off the Mackenzie Delta in the Canadian Beaufort Sea. The standard setup for the SWAN model was implemented by the authors, comparing different methods for quantifying the wave whitecapping dissipation. Hoque et al. [16] found that, after examining the bottom friction effects and triad interactions in predicting shallow-water waves, the simulated results in terms of significant wave heights and peak period are in good agreement with the observed data. Amunugama et al. [17] analyzed wind waves with SWAN on structured mesh and unstructured mesh during the arrival of a typhoon. After comparing the simulated results with measured data, they confirmed that the wind–wave characteristics obtained by both approaches (structured SWAN and unstructured SWAN) were substantially consistent with some advantages of unstructured SWAN, especially in complex geometries. The

authors recommended the combination of both approaches where necessary. Due to a lack of time-series wave data, Gorman et al. [19] have used SWAN and wind data to hindcast the generation and propagation of deep-water waves incident on the New Zealand coast over a 20-year period (1979–1998). The SWAN model was also used to analyze spatial and temporal variations in cold front events [20], predict waves generated by cyclones [21], simulate wave characteristics at shorelines [22,24], and estimate wave energy potential [23], etc.

This study aims at investigating the effect of extreme dredging depths (due to sand extraction) on the wave characteristics in the Oued Sebou estuary, located in the Kenitra region on the north Atlantic coast of Morocco by applying the SWAN model (Cycle III version 41.20). The wave hydrodynamic characteristics around the river outlet were examined. The dredging impact on wave dispersion, bottom friction velocity field, flow agitation in the river, and stress on the structures housing the estuary and wave energy budget (before breaking) were extensively analyzed.

## 2. Study Area

This study covers the Oued Sebou estuary area. The Sebou estuary is located on the Atlantic coast in the Kenitra region of Morocco in North Africa. It is a region where numerous coastal developments are located. Two concentrically spaced longitudinal breakwaters channel the Oued Sebou over a distance of about 800 m, extending to a bathymetric line of 7 m. The sandbar that prevents shipping in the canal was reduced by the construction of the breakwaters in 1931 (Figure 1). During low tide, these structures give the tidal current the ability (velocities and forces) to drive the sand along the shoreline in either direction. Additionally, this river is controlled by water agencies that guarantee its stability. The Mehdia beach (of the city side in the southern part of the river mouth) is surrounded by a corniche composed of a wall with an average height of 1.50 m and 2 m for the northern and southern halves, respectively. In the southern and northern beach zones, respectively, the wall's height above sea water level (SWL) is nearly 11 and 13 m. It is also important to note that the Mehdia shoreline dune is occupied by various manmade structures.

Dredging operations were carried out every year with an average volume of almost 460,000 $m^3$/year. This makes it possible to ensure adequate depth, which is necessary for the navigation of the fishing vessels and ships in direction of the Mehdia ports, located inside the channel (Figure 1).

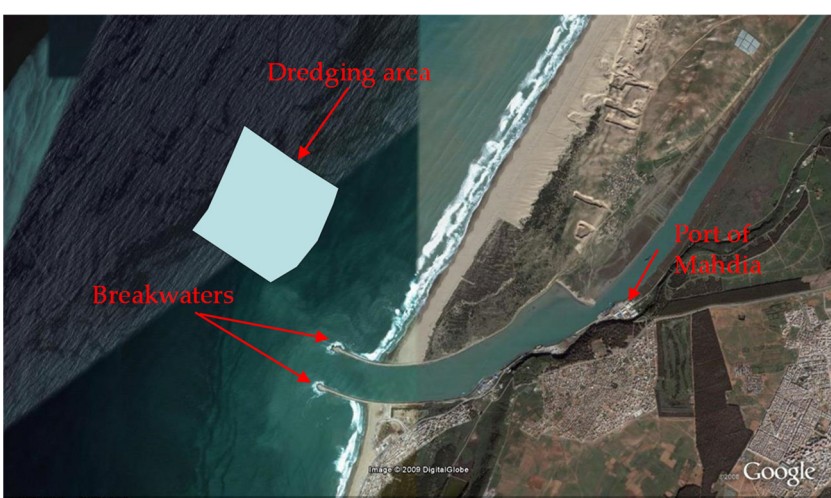

**Figure 1.** *Cont.*

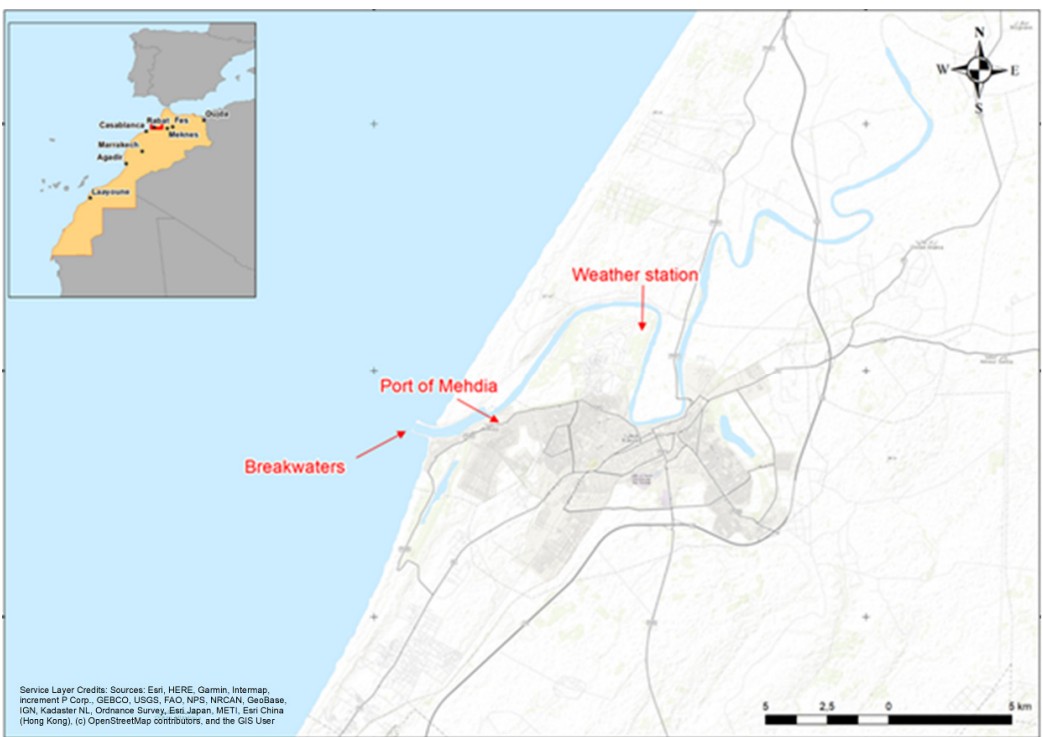

**Figure 1.** Outlet of the Oued Sebou into the Atlantic Ocean (above: Google Maps Platform) and a general overview of the study region (below).

### 3. Model Theoretical Formulation

The selected SWAN model is a spectral numerical model designed to simulate waves evolving in coastal regions, lakes, and estuaries under defined wind, bathymetry, and current conditions. It is based on the Energy Density Balance equation linking the advection term to the source and sink terms. Therefore, the wave energy evolves in both geographic and spectral space and changes its aspect due to the presence of wind at the surface, friction with the bottom, or during the breaking of the waves. The SWAN model is a stable model based on unconditionally stable numerical schemes (implicit schemes).

SWAN, in its third version, is in stationary mode (optionally non-stationary) and is formulated in Cartesian or spherical coordinates. The unconditional numerical stability of the SWAN model makes its application more effective in shallow water.

In SWAN, the waves are described with the two-dimensional spectrum of the wave action density $N$,

$$N(x, y, \sigma, \theta) = \frac{E(x, y, \sigma, \theta)}{\sigma} \tag{1}$$

where $x$ and $y$ are the horizontal components of geographic space, $\sigma$ is the relative frequency, $\theta$ is the wave direction, and $E$ is the energy density.

The spectrum considered in the SWAN model is that of the wave action density $N(\sigma, \theta)$ rather than the spectrum of the energy density $E(\sigma, \theta)$. This is because, in the presence of currents, the wave action density is conserved while the energy density is not [27].

Because wave action propagates in both geographic and spectral space under the influence of genesis and dissipation terms, wave characteristics are described in terms of two-dimensional wave action density. The action density spectrum balance equation relating the propagation term to the effects of the source and sink terms, in Cartesian coordinates, is [28]

$$\frac{\partial N}{\partial t} + \frac{\partial (C_x N)}{\partial x} + \frac{\partial (C_y N)}{\partial y} + \frac{\partial (C_\sigma N)}{\partial \sigma} + \frac{\partial (C_\theta N)}{\partial \theta} = \frac{S}{\sigma} \ . \tag{2}$$

On the left-hand side of Equation (2), the first term represents the local temporal variation of the wave action density, the second and third terms represent the propagation of wave action in the geographical space of velocities $C_x$ and $C_y$, the fourth term represents the shifting of the relative frequency due to variations in bathymetry (with propagation velocity $C_\sigma$) and currents (with propagation velocity $C_\theta$), and the fifth term represents the refraction induced by the combined effects of depth and currents. It is worth mentioning that $C_x$, $C_y$, $C_\sigma$, $C_\theta$ propagation velocities were obtained from linear wave theory. The term in the right-hand side of Equation (2) represents processes that generate, dissipate, or redistribute wave energy, and $S$ can be expressed as [24]

$$S = S_{in} + S_{wc} + S_{brk} + S_{bot} + S_{n14} + S_{n13} \, , \tag{3}$$

where $S_{in}$ is the wind energy input. The dissipation term of wave energy is represented by the contribution of three terms: the whitecapping $S_{wc}$, bottom friction $S_{bot}$, and depth-induced breaking $S_{brk}$. $S_{n14}$ and $S_{n13}$ represent quadruplet interaction and triad interactions, respectively.

Adopting a finite difference scheme for each of the five dimensions: time, geographic space, and spectral space made the numerical implementation in SWAN effective. The following guidelines must be followed in order to obtain the discretization adopted at the SWAN model level for numerical computation: (i) time of a constant and identical time step $\Delta t$ for the propagation term and the source term, (ii) geographical space of a rectangular grid with constant spatial steps $\Delta x$ and $\Delta y$, (iii) spectral space of a constant directional step $\Delta\theta$ and a constant relative frequency step $\Delta\sigma/\sigma$; (iv) frequencies between a fixed minimum maximum values of 0.04 Hz and 1 Hz, respectively, unlike the WAM and WAVEWATCHIII models, which use dynamic values, and (v) as an option, the direction $\theta$ can also be delimited by the minimum and maximum values $\theta_{\min}$ and $\theta_{\max}$.

## 4. Model Forcing Data

### 4.1. Bathymetry

The bathymetry of the study area is generated by using the SHOM charts, SHOM7702 SHOM Nautical Charts—Morocco. Available online: https://www.nauticalchartsonline.com/charts/SHOM/Morocco (accessed on 1 February 2021).

Figure 2a shows the bathymetry of the coarse grid domain, while Figure 2b illustrates a more detailed bathymetry. The seabed zone is characterized by comparatively regular isobaths parallel to the coastline. According to the bathymetry of the research region, shallow water extends up to 550 m from the coast with a depth range between 2 and 3 m.

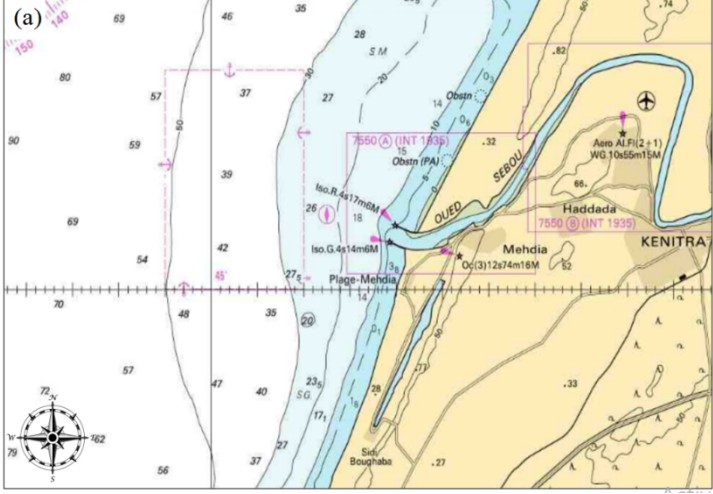

**Figure 2.** *Cont.*

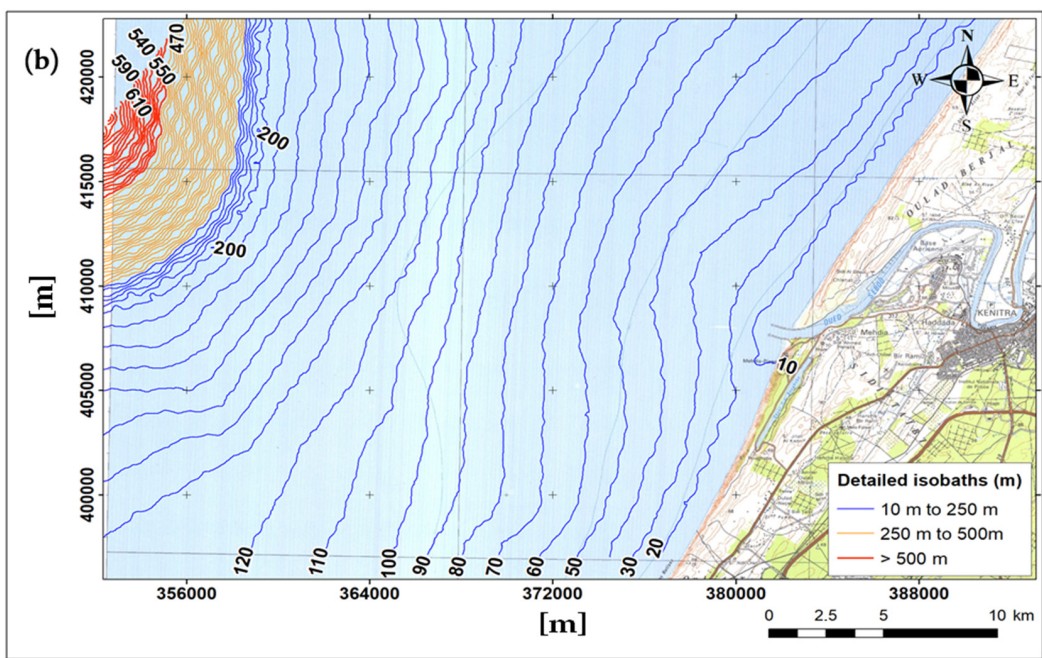

**Figure 2.** Bathymetry of Sebou estuary. (**a**) Bathymetry of the coarse grid domain. (**b**) Detailed isobaths of the study area.

Generally, for wave simulation, the research and commercial models use flexible grid mesh (structured and unstructured grids). In previous studies [17,21], it was observed that simulations with structured or unstructured grids were substantially consistent. In this study, we use a nested regular grid with a resolution of approximately 25 m, as shown in Figure 3. The bathymetry meshing was generated by using the BlueKenue software.

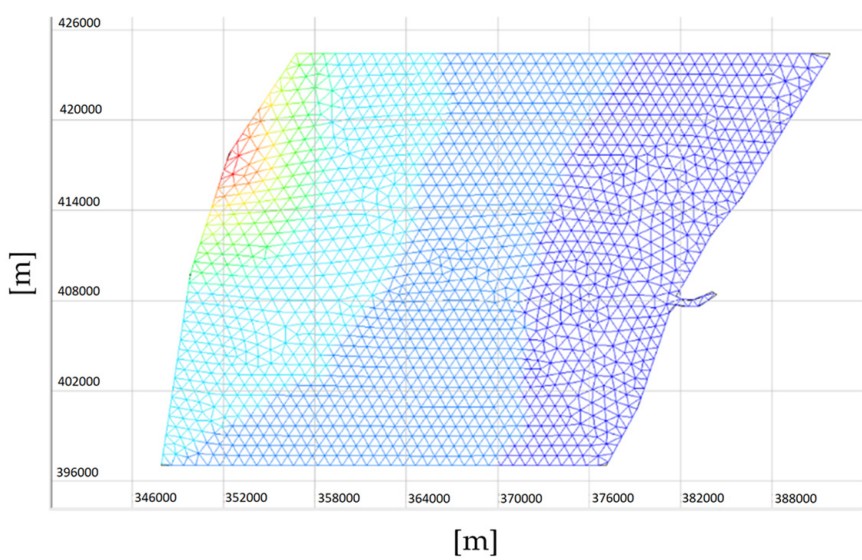

**Figure 3.** Mesh used of the study area: regular grid of approximately 25 m resolution.

### 4.2. Wind and Wave Fields

Because the accuracy of the wind field has a large impact on the predicted wave fields [16], in this study, the main meteorological parameters were well analyzed. The average annual values of the wind properties indicate that the study area is characterized by: (i) a winter regime for which the dominant wind (or most frequent) mainly comes from the eastern sector (onshore wind). During this period, the strongest wind speed (>9 m/s) comes from the directions ranging between the southwest and west sectors, with

an occurrence of almost 3%, (ii) a summer regime (from March to October), for which the dominant wind comes mainly from the western sector (sea wind). In this period, the strongest wind speed (>9 m/s) comes from the west and north sectors, with an occurrence of almost 3%.

The meteorological data (Figure 4) were collected from the nearest weather station to the study area. It is situated at the Kenitra airport, which is 8 km from the study location. The collected measured data correspond to the period from 1990 to 2009. The annual predominant wind direction is from the west and its speed range from 4 to 9 m/s (Figure 4c).

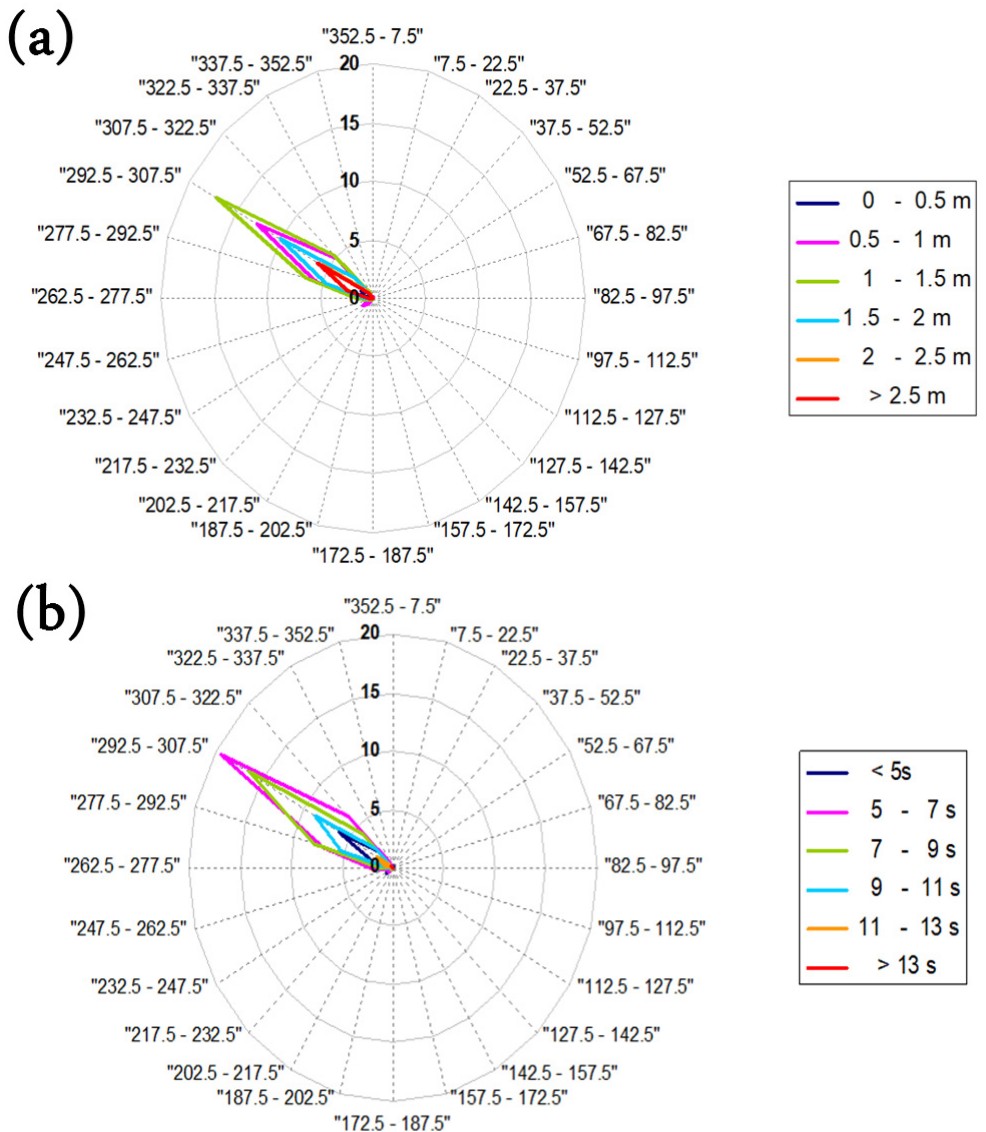

**Figure 4.** *Cont.*

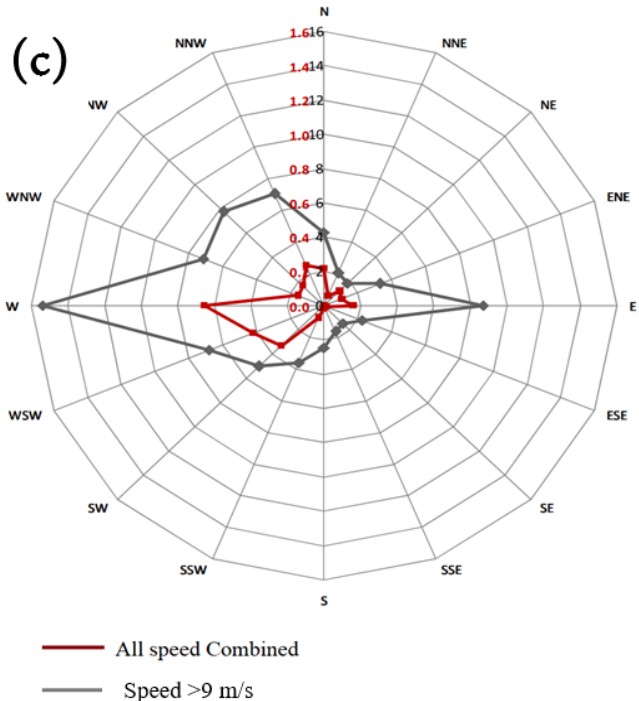

**Figure 4.** Wave rose (**a**) of the significant wave heights, (**b**) corresponding periods of the significant wave, (**c**) and average annual wind rose.

Figure 4 shows the annual wave rose of the significant wave heights (Figure 4a) and their corresponding periods (Figure 4b). The dominant wave (the most frequent) comes from the northwest (300 °N to 320 °N). Storm swells are more westerly and come from the west–northwest sector (280 °N to 300 °N). Five significant wave heights were considered in this study (results of 19-year data analysis), $H_{s0}$ = 1.5, 2, 2.5, 3, 3.5, and 4 m. Here, we denote by $H_{s0}$ the significant wave height in deep water.

*4.3. Tidal Current and Water Level*

The average sea level in the region of Kenitra is almost equal to 2.17 m. The tide is of semi-diurnal type with a period of 12 h 25 min. Previous studies confirmed that the tidal current and water level significantly affect wave behaviors [24]. As a result, in this study, the tidal forcing was also considered as SWAN mode input data l. Table 1 provides a summary of the astronomical tide level values for the study area.

**Table 1.** Tidal levels in the study area.

| Table | Low Tide (m) | High Tide (m) | Tidal Range (m) |
|---|---|---|---|
| Exceptional high water | 0.50 | 3.90 | 3.40 |
| Medium high water | 0.80 | 3.50 | 2.70 |
| M.M. (medium) | 1.40 | 3.00 | 1.60 |
| Medium still waters | 1.50 | 2.70 | 1.20 |

*4.4. Model Setup*

The site of the Oued Sebou estuary is exposed to the west–northwest direction, which is also the direction of the main dominant wave. Based on the layout of the coastline and the regularity of the hydrodynamic solicitations, the main sediment transport occurs along the channel profile. Monitoring of the estuary area shows that there is no significant littoral transit.

Regarding the primary swell direction (N300), the most exposed areas to wave ac-tion are located downstream of the river. These locations have been given the labels A, B, C, and D, as shown in Figure 5.

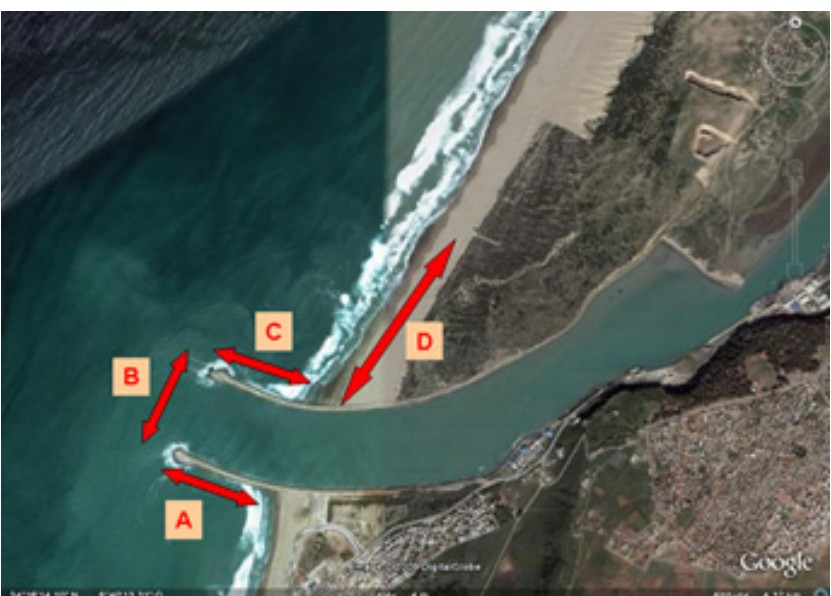

**Figure 5.** The different zones subject to analysis: stress on structures (A, C), B-agitation in the river outlet (B), littoral erosion (D).

After considering the wave and tidal conditions, we chose to proceed with a selection of eight (8) modeling cases (see Table 2), combining the considered five significant wave heights, indicated above, and different tide levels ranging between 0 and 3.4 m. The selection of wave characteristics covers the most representative samples. The current bathymetry (without dredging), the bathymetry with a dredged area of uniform extraction depth of 2 m (with respect to the current bathymetry), and the dredged area of uniform extraction depth of 4 m were all taken into consideration. Moreover, the SWAN model was used with the following additional hydrodynamic parameters: energy spectral density distribution (JONSWAP) of a spectrum width parameter at 3.3 and a low angular spread, spectrum discretization of 36 angular sectors and 32 frequency intervals (between 4 and 20 s), the refraction and diffraction phenomena were considered, and the wind turbulence effects were not considered.

**Table 2.** Specification of the hydrodynamic cases considered.

| | Wave | | Tide |
|---|---|---|---|
| Case Ref. | Significant Height $Hs_0$ (s) | Period $T_p$ (m) | Level (m) |
| 1 | 4.0 | 12 | 0.0 (low tide) |
| 2 | 2.0 | 10 | 0.0 (low tide) |
| 3 | 1.5 | 8 | 0.0 (low tide) |
| 4 | 1.5 | 6 | 0.0 (low tide) |
| 5 | 4.0 | 12 | 2.2 (intermediate level) |
| 6 | 2.0 | 10 | 2.2 (intermediate level) |
| 7 | 1.5 | 8 | 2.2 (intermediate level) |
| 8 | 4.0 | 12 | 3.8 (high tide) |

## 5. Results and Discussion

As an illustration, Figure 6 depicts the simulated outcomes of the local significant wave height ($H_s$) distribution, before (with current bathymetry), as shown in Figure 6a, and

after dredging of 2 m, as shown in Figure 6b. The data illustrated in Figure 6 refer to case 3 (Table 2) of significant wave height in deep water $H_{s0}$ = 1.5 m and a period $T_p$ = 8 s.

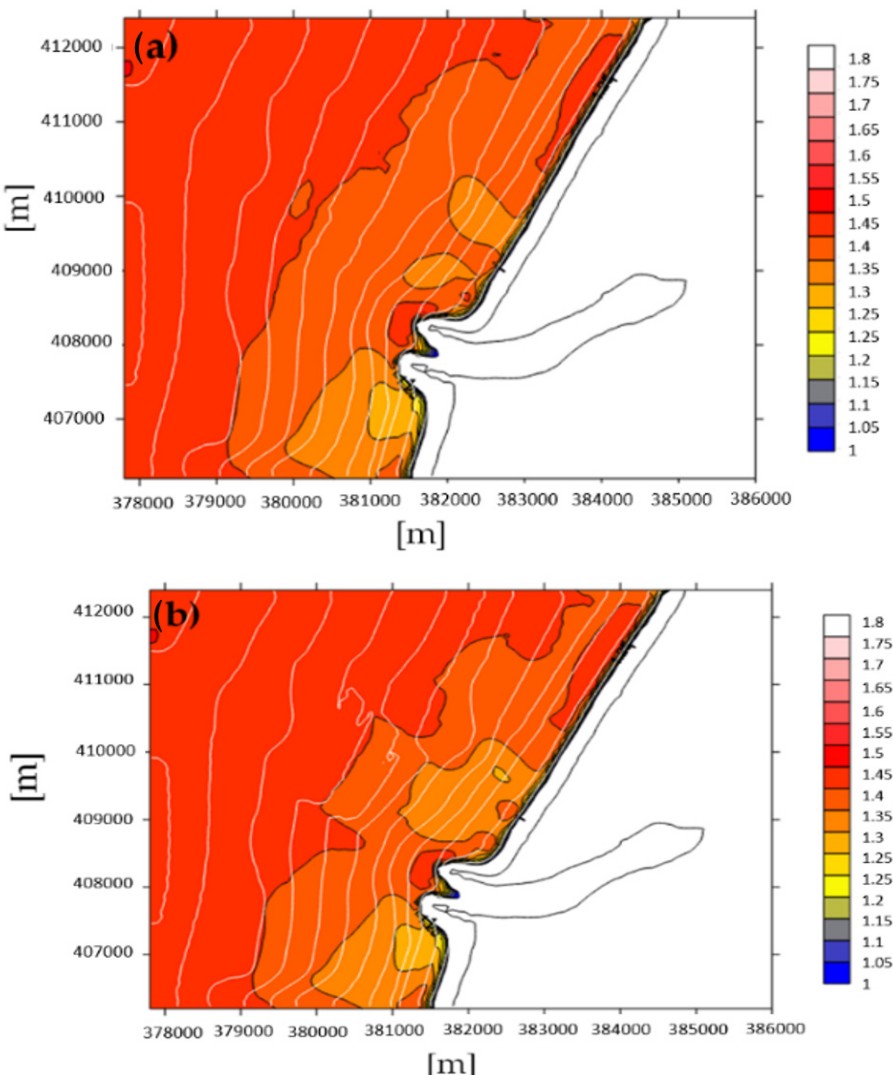

**Figure 6.** Map of local significant wave height distribution for ($H_{s0}$ = 1.5 m, $T_p$ = 8 s), (**a**) before dredging (with current bathymetry), (**b**) after dredging of 2 m depth. The legend indicates the values of $H_s$ in meters.

The results show that the presence of the dredging affects the wave dispersion. There was an increase in $H_s$ in and at the edges of the dredging area, which can be explained by the sudden increase in the water depth. In general, the simulations show a decrease in the transfer of wave energy out of the extraction zone, causing a lateral energy flow with respect to the excavation. The waves propagating across the excavation area tend to refract toward the areas of shallow water along the edges, increasing the wave heights (known as wave energy focusing). In terms of wave amplitude, there is a modest decrease downstream (in the wave propagation direction) of the excavation and a slight increase inside it and on its sides. This behavior is also confirmed with extreme waves, as shown in Figure 7. In Figure 7, the data of $H_s$ refer to case 8 (Table 2) of $H_{s0}$ = 4 m and a period $T_p$ = 12 s, simulated with and without the dredging of 2 m depth.

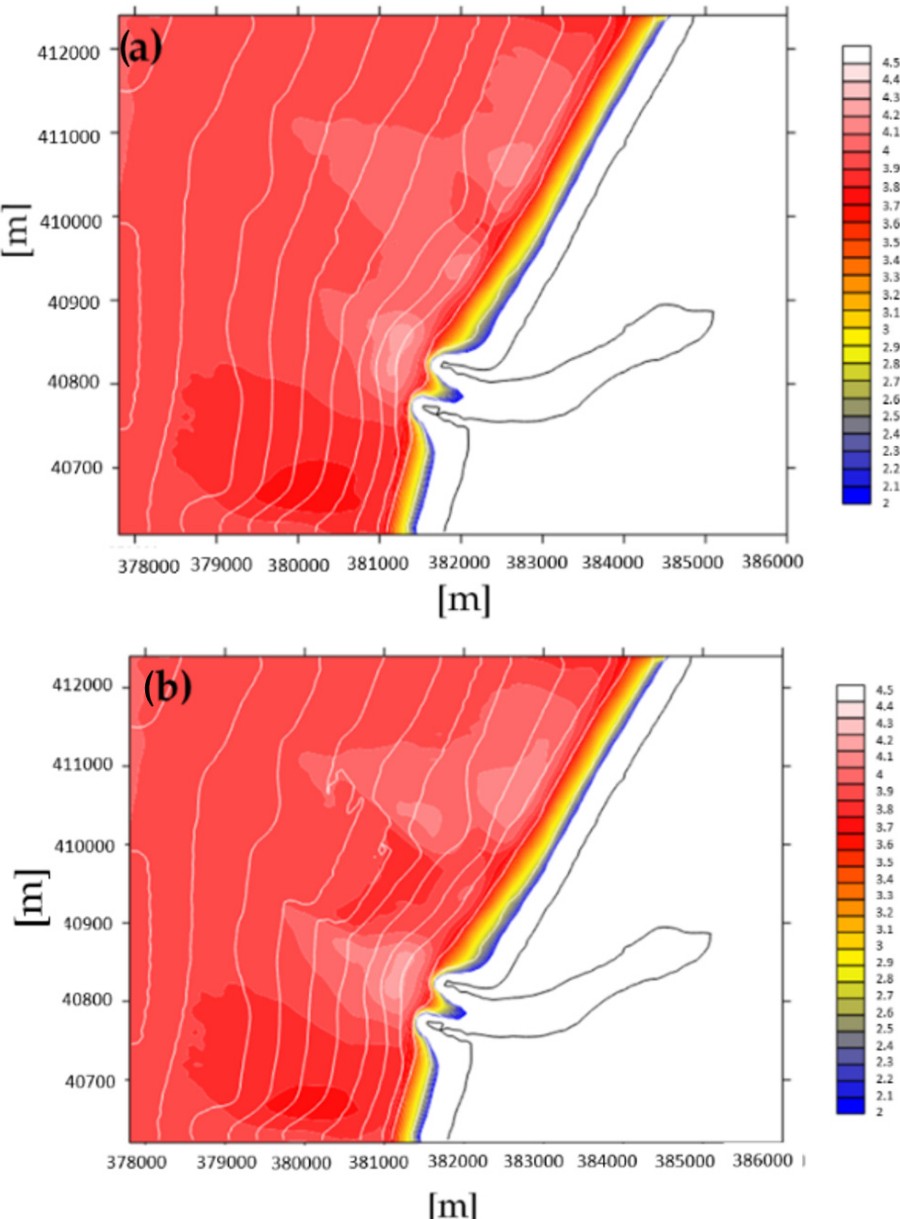

**Figure 7.** Map of local significant wave height distribution for ($H_{s0}$ = 4 m, $T_p$ = 12 s), (**a**) before dredging, (**b**) after dredging of 2 m depth. The legend indicates the values of $H_s$ in meters.

To evaluate the impact of dredging on the estuary environment, two indicators have been considered: (i) impact on sediment transport processes, where the analysis focuses on the evolution of the bottom friction velocities, responsible for the movement of sediment particles and proportional (velocity squared) to the erosion rate, and (ii) impact on structures, where the analysis focuses on the variation of the energy before the wave breaks, at each concerned zone (Figure 5). For this final point, the impact of the significant heights (squared), measured at the bathymetry of 10 m, and on each of the four selected areas was integrated. In practice, the values of $H_s^2$ determined at different locations $S_A$, $S_B$, $S_C$, and $S_D$ (Figure 8), along the isobath of 10 m, were performed to indicate the wave energy impact on the selected zones, as shown in Figure 5. A comparison was made between relative configurations with and without dredging.

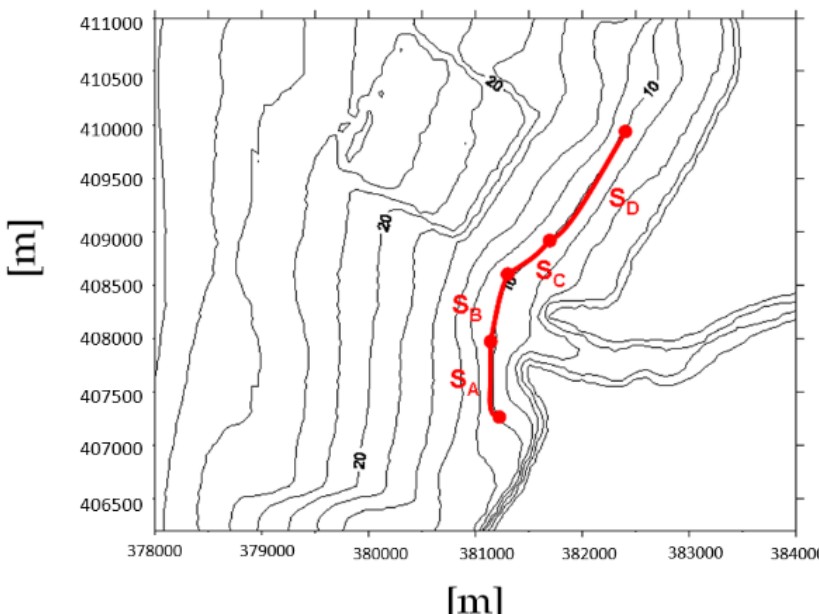

**Figure 8.** Segmentation for the analysis of energy flows in the concerned zone. The contour lines indicate the bathymetry values.

Figure 9 depicts examples of the relative bottom friction velocity (reflecting the bottom resistance) distribution in the target area. The relative friction velocity is defined as the difference between the simulated friction velocity with bathymetry with the dredged area and that without dredging (current bathymetry). The data shown in Figure 9 refer to the dredging of 2 m depth. Figure 9a shows the results of the relative friction velocity obtained with the most frequent case ($H_{s0}$ = 1.5 m, $T_p$ = 8 s) (see Table 2), whereas Figure 9b shows that of the highest wave condition ($H_{s0}$ = 4 m, $T_p$ = 12 s). Figure 9 shows the clear effects of the dredging on the bottom friction velocity distribution.

The bottom friction velocity changes significantly as the wave gets closer to the excavation site. Locally, around the borders of the excavation, a minor increase of bottom resistance is noted, as demonstrated by the positive relative friction velocity values in Figure 9. The bottom friction velocity significantly decreases inside the dredging area, reaching maximum magnitudes (of negative values). The effect of the dredging area on the friction velocities extends downstream of it, reaching zones C and D (Figure 5).

The variation of the bottom friction velocity due to the excavation certainly affects the sediment transport potential [29–31] and the wave energy flux linked to the redirected waves, particularly in zones C and D. Figure 9 indicates a clear increase in the bottom friction velocity at the lateral sides (in the wave direction) of the excavation, which is more pronounced with the high tide condition ($H_{s0}$ = 1.5 m, $T_p$ = 8 s). The possibility for erosion action increases as bottom shear stress increase. The friction velocity at the sea bottom slightly decreases in zones C and D, which are located downstream of the extraction site. This suggests that zones C and D are likely subject to sediment accumulation following dredging activities. An excessive sediment buildup can cause several environmental problems. It can reduce the seawater depth, preventing the passage of ships. It can also lead to contamination that poses a threat to aquatic plants (*Posidonia oceanica*) and wildlife. Zones A and B of the coastal area in the southern part (from breakwaters) are almost not affected by the excavation site. Additionally, findings indicate that dredging up to 2 m depth has no significant effect along the channel profile.

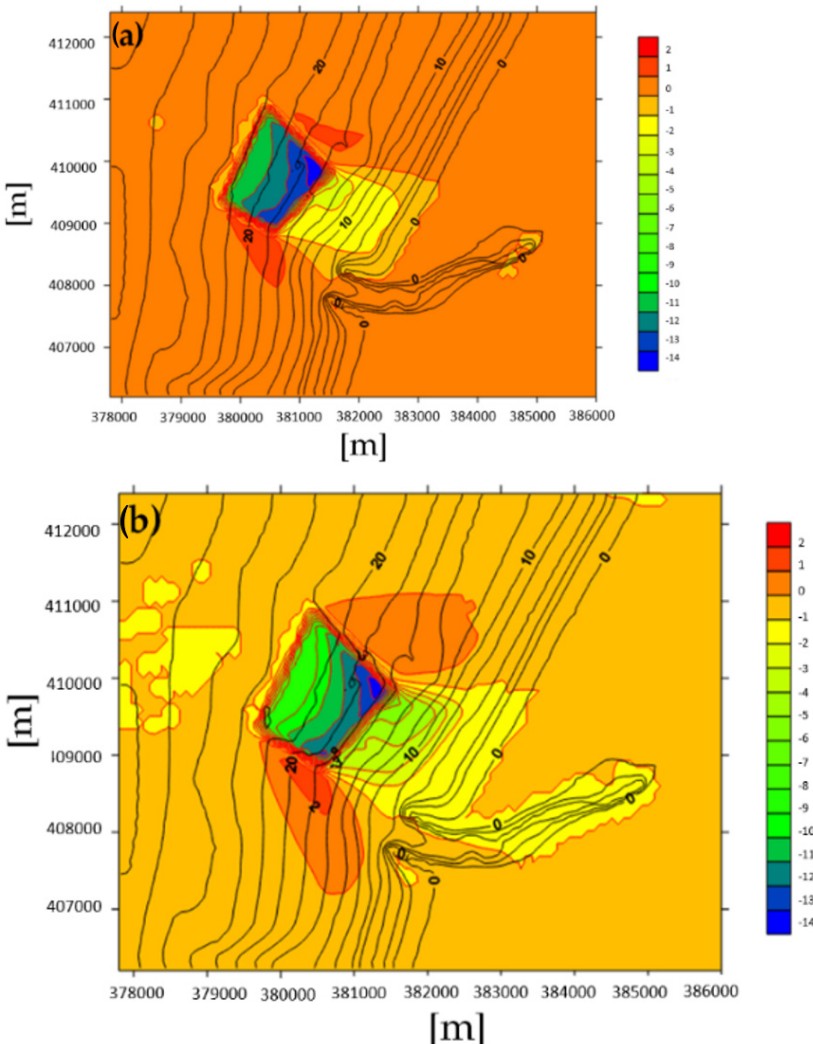

**Figure 9.** Relative bottom friction velocity, (**a**) ($H_{s0}$ = 1.5 m, $T_p$ = 8 s), (**b**) ($H_{s0}$ = 4 m, $T_p$ = 12 s) after dredging of 2 m depth. The legend indicates the values of relative friction velocity in [cm/s]. The contour lines indicate the bathymetry values.

Figure 10 shows the wave height, $H_s$ (Figure 10a) and its relative velocity (Figure 10b) distribution maps, with additional excavations up to 4 m deep. Figure 10a demonstrates how waves propagating across the excavation area tend to refract toward the areas of shallow water along the edges, increasing the wave heights (known as wave energy focusing). This causes the amplification of $H_s$ at the level of the river mouth, affecting zone B (Figure 5). The increase in wave height at the mouth of the river continues to propagate upstream along the channel. As a kind of obstruction, the excavation causes the incident waves to converge and creates a wake region downstream of it where the wave height decreases.

Similar behavior to the wave height distribution is shown in Figure 10b, where the relative friction velocity oscillates near the bottom. The friction velocities increase along the excavation's lateral edges and decrease downstream of it. Compared to Figure 9a, dredging up to 4 m deep has a greater impact on the estuary environment than dredging up to 2 m deep. With 4 m of dredging, the area is subject to increased erosion. Zones C and D, however, are subject to sediment accumulation.

Estuary ecosystems are significantly impacted by wave energy. The bathymetry in some coastal regions causes a concentration of wave energy, which raises wave height [32]. Figure 11 displays the fraction of wave relative energy percentage that corresponds to zones A, B, C, and D (Figure 5). The energy was estimated at 10 m isobath, as shown in Figure 8,

and the relative percentage of energy was determined as the difference between the energy of the simulated wave with the excavation present and without it. The percentage is in relation to the simulated wave energy without excavation. The data shown in Figure 11 relates to a 2 m-deep excavation.

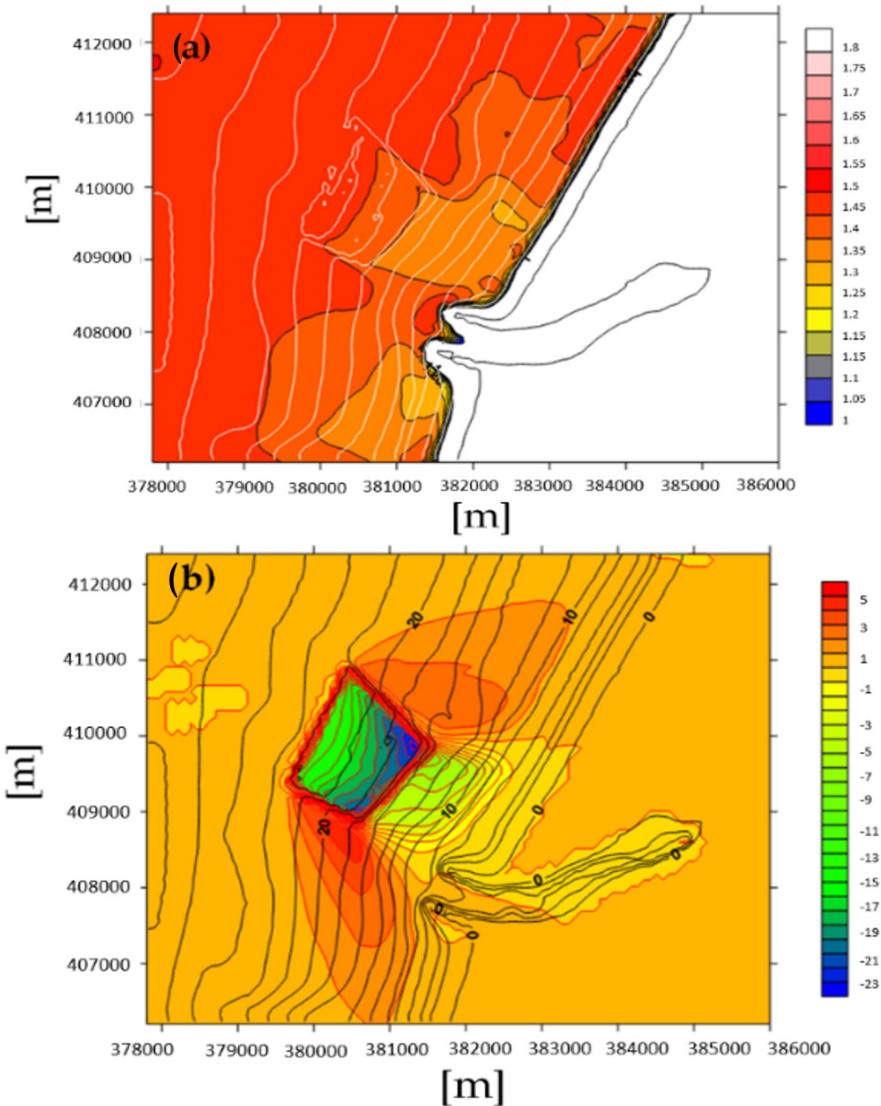

**Figure 10.** Impact of dredging 4 m-deep with ($H_{s0}$ = 1.5 m, $T_p$ = 8 s) (**a**) significant wave height distribution after dredging, (**b**) relative friction velocity at the seabed after dredging up to 2 m depth. The legends indicate the values of $H_s$ in meters and those of relative friction velocity in [cm/s]. The contour lines indicate the bathymetry values.

Figure 11 demonstrates that the cases 1, 5, and 8 (see Table 2), with the largest values of $H_{s0}$ (4 m), undergo the greatest values of energy variations. The condition of the lowest sea level (case 1), in the different zones from A to D, consistently exhibits the highest variation in energy. The smallest variations in energy always appear in cases 4 and 7, with the lowest values of $H_{s0}$ (1.5 m). Figure 11 further demonstrates that zone A, which has a maximum variation value of order 4%, is the most affected zone by wave energy. The energy impact gradually decreases in the order from zone A to zone D, which is especially prominent in zones C and D. In general, it can be concluded that the dredging activities have a minimal but noticeable impact on the estuary environment of Oued Sebou that is not significant.

The simulated results of wave dispersions and their hydrodynamic structures are useful for estimating the sediment transport rates that will be an extension of this work.

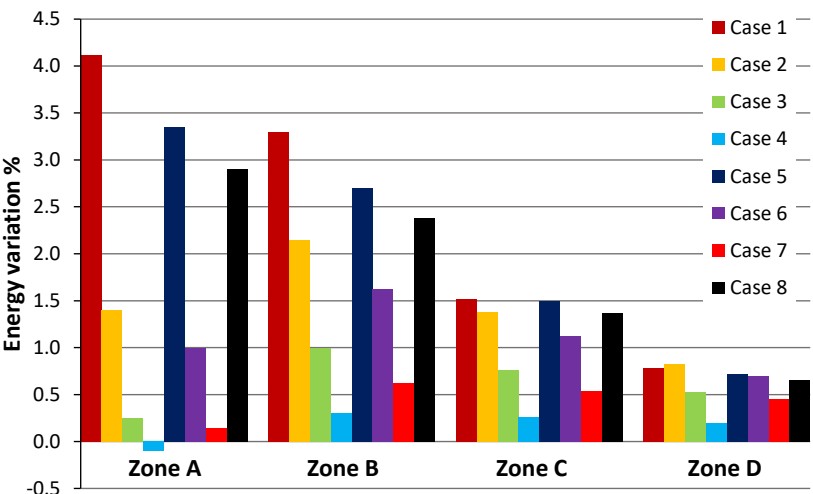

**Figure 11.** Energy percentage change in zones A, B, C and D.

## 6. Conclusions

The SWAN model was used to simulate a large number of wave motions at the Kenitra site in order to better understand the effects of dredging activities (due to sand extraction) in the Oued Sebou estuary. In various meteorological conditions, comparisons between the simulated results with and without dredging bathymetries were made. Eight configurations that accurately reflected the actual hydrodynamic circumstances that characterize the study area were tested.

The effects of the bathymetric changes, due to the sand extraction, on wave dispersion, bottom friction velocity field, and energy budget (before breaking) were extensively analyzed.

The sharp variation of the bathymetry due to the dredging that increases water depth causes an increase of the local significant wave height in and at the edges of the area susceptible to sand extraction. According to the simulation results, the waves propagating across the excavation area tend to refract toward the areas of shallow water along the edges, increasing the wave heights (known as wave energy focusing). This causes the amplification of $H_s$ at the river mouth level and along the southern breakwater structure, which is more pronounced with deeper dredging. The excavation plays the role of a kind of obstacle, causing the incident waves to converge and creates a sort of wake region downstream of it where the wave height decreases.

The results show that zones C (at the northern breakwater) and D (northern coastal area) are subject to possible accumulation of sediments, whereas zones A (at the southern breakwater) and B (river mouth) are subject to an increased potential for erosive action and a risk of scouring processes at the southern breakwater.

Dredging activities in the Oued Sebou estuary mainly affect the river mouth (zone B) and the southern breakwater area (zone A), which is very noticeable with dredging up to 4 m deep.

In general, it can be concluded that the dredging activities show a certain level of impact on the estuary environment of Oued Sebou that is not very significant.

Despite the crucial role dredging plays in the nation's economy and maritime engineering management (i.e., it helps make the water navigable, removes contaminants from seabeds and recreates damaged areas, maintains many marine infrastructures, and many other advantages), dredging could have serious and long-lasting negative impacts on the environment, leading to contamination that poses a threat to aquatic plants (*Posidonia oceanica*) and wildlife

The simulation results, which will be validated by some measured wave characteristic data, are useful for examining sediment variations along the estuary coastal area, a subject we are currently working on.

**Author Contributions:** N.I. and M.C., performed the numerical simulations and methodology; M.B.M. and N.I. formal analysis, study design, writing–original draft preparation; L.M., M.C. and D.L. contributed suggestions, discussions, and reviewed the manuscript. All authors have read and agreed to the published version of the manuscript.

**Funding:** This research received no external funding.

**Data Availability Statement:** The data presented in this study are available on request from the first author (Nisrine Iouzzi).

**Acknowledgments:** This study was carried out at Hassan II University of Casablanca, Faculty of Sciences Ben M'Sik, (Morocco).

**Conflicts of Interest:** The authors declare that they have no conflict of interest.

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
