# Peer review of "SWAN Modeling of Dredging Effect on the Oued Sebou Estuary"

_water, doi:10.3390/w14172633_

Round 1

Reviewer 1 Report

Manuscript ID: water-1840700

Manuscript title: SWAN Modeling of dredging effect on the Oued Sebou Estuary

Review report by Antonio Mubango Hoguane

Date: 1st August 2022

Overview and general recommendation

Dredging plays a vital role in the nation's economy since it involves, among others, the extraction of valuable mineral resources and enabling navigation of commercial cargo ships. However, dredging could have serious and long lasting negative impacts on the environment. For example, it can damage or destroy fish spawning grounds, change on wave characteristics with consequence on erosion and sedimentation processes. Hence, the present study, on investigating the effect of dredging the naturally build-up of sand and silt at the Oued Sebou estuary mouth for navigation purpose, is of great importance for both science and management purposes.

The article is very relevant to Morocco, and may well be of interest to wider scientists working in estuaries and coastal waters. Further, the study fits within the sub-section: Hydraulics and Hydrodynamics, and so within the scope of the journal.

I recommend the manuscript to be accepted for publication with major revision. Bellow, I present some suggestions for improvement of the manuscript, would the authors want to consider.

In general, the manuscript requires some English improvement. Further, I suggest the authors to check the structure. For instance the model description should be immediately followed by model step-up and model forcing and validation. In fact, it is advisable that these be part of the same section. This may imply shifting the study area description to a position immediately after the introduction section. Perhaps, the authors may want to consider a standard structure of papers which is: Abstract, Introduction, Methodology, Result, Discussion, and Conclusion, where study site and model descriptions may fall under Methodology and Result and Discussion may be clustered together.  

Abstract

Given the importance of the article in science and management, I wish the authors would have given an informative abstract, and in such a case the present abstract it is lacking results and conclusions.

Study area

Location map: It would be helpful if the location map is put into a geographical perspective, i.e. showing which region of Africa it is located. Please note that Figure 1 does not show the scale to give the reader the idea of the size of the study area. Further, if the location map is given on GIS map rather than google map, it would be better. Furthermore, relevant places mentioned in the report, such as the meteorological station at Kenitra airport and Kenitra area should be indicated in the study area Figure 1. This is shown in Figure 2 but it should be shown in Figure 1 as well.

Result

The result section also discusses the result… I wonder if it cannot be termed “Result and Discussion”. Further, the discussion should also be directed to give insight of the meaning of the result for environmental management and for engineering, apart from discussing the causes of changes. Lines 311-317, in the Result section and lines 386-394, in the Conclusion section, give a brief indication of the implications of dredging on environment. However, I wish the authors had explored/discussed furthermore these implications throughout the result section.

It is advisable that Figure 9a and Figure 9b be on same colour scale. Similarly to Figure 10a and 10b.

_____________________________________

Reviewer 2 Report

The present study utilizes SWAN model to study the impact of dredging on the hydrodynamic in the Oued Sebou estuary. Different case studies are carried out to do the sensitive analysis. two locations at different depths are selected to study the impact in the wave parameters near the estuary. The flow of information in the paper is very smooth defining each aspect very briefly. The authors have discussed every test cases with proper detailed explanation. The conclusions are well written. This paper forms a good piece of information related to dredging activities as it will be beneficial to the coastal communities. It may be accepted for publication with major revision.  

The following points may be considered for review:

1.      Language should be checked thoroughly at many places throughout the manuscript, especially grammer.

2.      Abstract: Introductory lines in starting of the abstract can be reduced. Significant results obtained from the study can be included in 1-2 lines.

3.      Ln98: Avoid repetition of the citation.

4.      Ln112:define the model version number. Give reference

5.      Figure 3 caption can be reframed. Instead of ‘Meshing’ use ‘Mesh used’

6.      Ln 204: Mention the parameters. If the data is measured data then the accuracy is correct. How this statement is applicable here? If study is using any reanalysed data then the accuracy of the data has to be assessed and it has to be validated with the measured data. The sentence can be deleted or reframed w.r.t content.

7.      Ln206-212: Give rose plots for wind data with seasons as explained in this para. It will give better understanding for readers.

8.      Ln 213: Specify the data. Also mark the weather station in the study region/bathymetry figure. Also give duration of the data used. Any previous studies describing winds in this region? If yes please cite them.

9.      Ln 216-220:Explain Figure 4 in more details in regards with wave height (seas and swells) and periods. Also use constant values of direction in the rose plots instead of giving range as the area has one dominant direction, so giving range for each quadrant is of no use. Also specify direction for each 4 quadrants. It is easy to understand for readers.

10.  Ln 218-219: This statement is not clear. Are these initial conditions or boundary values? Whether SWAN model was setup using wind input for whole duration of the study? Be specific with this details.

11.  Ln 224: Expand ‘NGM’

12.  Table 1: units are missing

13.  Ln 231-232: Does it mean that the dominant wave direction is west/northwest? If not then specify the dominant wave direction.

14.  Ln 235-237: These lines can be shifted to last para of Introduction as this section discusses various inputs given to the model.

15.  Ln 242: specify what are the wave types considered in this study.

16.  Ln 248: what are the total inputs given to the model? What are other physics used in model setup? Whether winds were given. It is better to rename this section as model setup and write all the details more clearly.

17.  Table 2: Put significant wave height as first column and period as second column as in further discussion authors have followed this sequence. Expand Tpic in the text. Peak period is usually denoted as Tp.

18.  Section 5: Rename as Results and Discussion.

19.  In all figures increase the font size. Values are not visible.

20.  Ln 293 –in this paragraph author has suddenly discussed about relative current velocity distribution. Nowhere in the manuscript is mentioned about this.

21.  Ln 301:How do the authors gets this output (bottom current velocity)? SWAN model will give wave parameter and spectra. Authors may specify details about the same in methodology. Also authors should be specific about the velocities, do they refer it to velocity components? as the values are in negative it cannot be current speed. Also why current directions are not discussed?

22.  In fig 11, zone A we see negative variation in Case 4. Justify that variation.

23.  Conclusions: Ln 369: About sand extraction can also be mentioned in initial part of the manuscript.

24.  Ln 379-381: This is also mentioned first time in conclusions. Link it somewhere in results and discussion.

Round 2

Reviewer 2 Report

The authors have incorporated all the suggestions and modified the manuscript. Very important: the authors should improve the quality of all the figures. Axis values are not at all visible.

Author Response

According to your suggestion, most of the figures have improved. Figures 3-10 have been greatly improved by redesigning them and increasing their size to make the axis values more visible.